# Joint Angle Variability Is Altered in Patients with Peripheral Artery Disease after Six Months of Exercise Intervention

**DOI:** 10.3390/e24101422

**Published:** 2022-10-06

**Authors:** Farahnaz Fallahtafti, Zahra Salamifar, Mahdi Hassan, Hafizur Rahman, Iraklis Pipinos, Sara A. Myers

**Affiliations:** 1Department of Biomechanics, University of Nebraska at Omaha, Omaha, NE 6160, USA; 2Department of Surgery and VA Research Service, VA Nebraska-Western Iowa Health Care System, Omaha, NE 68105, USA; 3School of Podiatric Medicine, University of Texas Rio Grande Valley, Harlingen, TX 78550, USA; 4Department of Surgery, University of Nebraska Medical Center, Omaha, NE 68105, USA

**Keywords:** claudication, walking performance, vascular disease, arterial disease, joint kinetics

## Abstract

Supervised exercise therapy (SET) is a conservative non-operative treatment strategy for improving walking performance in patients with peripheral artery disease (PAD). Gait variability is altered in patients with PAD, but the effect of SET on gait variability is unknown. Forty-three claudicating patients with PAD underwent gait analysis before and immediately after a 6-month SET program. Nonlinear gait variability was assessed using sample entropy, and the largest Lyapunov exponent of the ankle, knee, and hip joint angle time series. Linear mean and variability of the range of motion time series for these three joint angles were also calculated. Two-factor repeated measure analysis of variance determined the effect of the intervention and joint location on linear and nonlinear dependent variables. After SET, walking regularity decreased, while the stability remained unaffected. Ankle nonlinear variability had increased values compared with the knee and hip joints. Linear measures did not change following SET, except for knee angle, in which the magnitude of variations increased after the intervention. A six-month SET program produced changes in gait variability toward the direction of healthy controls, which indicates that in general, SET improved walking performance in individuals with PAD.

## 1. Introduction

Peripheral artery disease (PAD) results from systemic atherosclerosis and leads to the obstruction to the arteries that supply blood to the legs. More than 200 million people worldwide are affected by the PAD, which leads to poor health condition, sedentary lifestyle, and physical dependence [1,2,3]. In Western countries, the prevalence of PAD is around 13% among people over 50 years of age [4]. The primary symptoms of PAD are intermittent claudication and pain during walking due to the oxygen supply to the leg muscles not meeting metabolic demand [4,5]. This causes severe impairments during walking and limitations in mobility [4,5,6]. Most patients with PAD are elderly, have mobility restrictions, and experience poor health outcomes, including an increased risk of falls [7,8,9,10].

Altered gait in patients with PAD is characterized by the decreased ankle, knee, and hip muscle power contributions and weakness of the ankle and hip muscles [11]. A consistent finding is reduced ankle muscle contribution and resulting power output, which decreases overall energy output during push-off [5]. As a result, walking efficiency is reduced in these patients compared to healthy individuals [12,13]. There is a common belief that patients with PAD have mobility impairments after the onset of intermittent claudication. However, previous work determined that patients with PAD have significant gait changes from the first step taken, prior to pain onset. These changes occur in both gait biomechanics and gait variability studies [3,10,14]. 

Supervised exercise therapy (SET) is the primary conservative treatment for patients with PAD [15,16]. SET has been well documented in its ability to increase the distances patients can walk and in improving quality of life [11]. Guidelines for effective SET programs include being conducted continuously, for at least four months, consist of three sessions per week lasting 30 min or more, and with an intensity that encourages patients to walk until near the maximal threshold [11]. However, only limited improvement in gait parameters have been reported after the SET [11,17].

Gait variability parameters provide insights into the coordination between all components involved in locomotor control of movement, and it is indicative of altered locomotor control [3,10]. Analyzing the variability of movement patterns in patients with PAD can also provide insight into movement control, allowing for the development of appropriate prognostics and diagnostic tools [10]. Patients with PAD have increased amount and more irregular structure of gait during walking compared to healthy young and older adults, respectively [3,10]. There is an association between changes in gait variability and a higher risk of falling [10]. Specifically, increased lower limb variability is associated with an increased risk of falls, especially if the patient is unable to produce sufficient compensatory response [18]. Patients with more irregular or random gait have less control during each step than patients who have an optimal level of gait variability [19]. How specific gait alterations result in gait variability differences from the “optimal” levels has not been established. 

Neuromuscular control of proximal and distal joints may affect the coordination of movement patterns [20]. In terms of motor learning that occurs when practicing a task, the proximal joints become under control prior to the distal joints [21]. Prior studies have found an increased variability and irregularity of movement at the ankle compared to the hip in healthy young individuals [19,20,21]. Movement patterns of proximal segments (i.e., trunk) had more organized patterns compared to distal segments (shank and foot) during walking in both younger and older adults [22]. These differences may relate to greater inertia of the proximal segments compared to the distal foot and shank. In another study, harnesses worn during walking were reported to negatively affect ankle neuromuscular control (i.e., decreased organization of variability patterns) compared to the knee and hip [23]. In healthy young adults, joint angle variability during continuous stair mill climbing were similar across lower limb joints [24]. With known calf muscle impairments, which impact ankle contributions to walking in patients with PAD, levels of variability could differ across joints. Walking exercise that is part of the SET intervention may improve muscle strength, coordination, and variability patterns.

Thus, this study aimed to determine the effect of a 6 month SET intervention on gait variability in patients with PAD. We used measures of amount (linear) and structure (nonlinear) of gait variability to identify the changes in lower limb joint angles during walking. We expect to observe a significant effect of the intervention on dependent variables. Dependent variables include sample entropy, largest Lyapunov exponent (nonlinear measures), standard deviation, and range of motion (linear measures) of joint angles in the sagittal plane.

We hypothesized to observe reduced sample entropy, largest Lyapunov exponent, and standard deviation of joint angles after the intervention while the range of motion increases. We believed the values would be reduced in a manner that made variability in patients with PAD more similar to those previously identified in healthy controls.

## 2. Materials and Methods

### 2.1. Participants

Forty-three patients with PAD were recruited from the vascular surgery clinics of Nebraska Western Iowa Veteran Affairs Medical Center and University of Nebraska Medical Center. The Internal Review Boards of the Nebraska Western Iowa Veteran Affairs and the University of Medical Centers approved the study. All subjects provided informed consent prior to participation in the study. The study was conducted in accordance with the Declaration of Helsinki, and the protocol was approved by the Ethics Committee of IRB #1576204. 

The Inclusion criteria of this study were: (1) having a history of chronic claudication, (2) exercise-limitation based on history or direct surgeon observation, (3) having no history of revascularization, (4) an ankle-brachial index of <0.90 at rest, (5) without musculoskeletal or neurological symptoms other than PAD that limit walking. The exclusion criteria include (1) not having PAD, (2) rest pain or tissue pain because of PAD, or (3) exercise limitations due to joint, musculoskeletal, neurological, heart, or lung disease rather than claudication.

### 2.2. Supervised Exercise Training Protocol

Based on previous research and the American College of Sports Medicine recommendations, each subject participated in a six-month, three-times-per-week SET program [11,25,26,27]. Subjects were supervised in the Center of Cardiovascular Disease Prevention and Rehabilitation of Creighton University Medical Center or associated rehabilitation centers in Nebraska and Western Iowa under a unified protocol. Each SET session contained 5 min warm-up, 50 min of intermittent exercise, and 5 min cool down. Treadmill walking exercise started at a low treadmill workload of 2 mph and 0% grade. Subjects walked on the treadmill until moderately severe claudication pain occurred, then stepped off the treadmill and rested until the pain stopped. This was repeated for a total of 50 min. As soon as a patient was able to walk for 8 to 10 min at the initial load, the grade and the speed were increased by 1–2% and/or by 0.5 mph. 

### 2.3. Experimental Data Collection

All the data were collected at the Biomechanics Research Building at the University of Nebraska at Omaha, before and after the 6-month SET program. Subjects were allowed to get familiar with treadmill walking and to select a comfortable speed, which was used during the testing condition. Thirty-three retro-reflective markers were placed on each subject’s lower extremity anatomical location, utilizing the marker system of Vaughan and Nigg [28,29]. Gait kinematics were recorded using twelve high-speed infrared camera systems (60 Hz, Motion Analysis Corporation, Rohnert Park, CA, USA). Gait variability analysis was performed on data from walking prior to the onset of claudication pain while the treadmill was set at 0%. 

### 2.4. Data Analysis

A custom MATLAB code was designed to process the marker coordinate trajectories. To obtain the most accurate results, the data were analyzed without filters. The relative joint angle time series from the lower extremity kinematic data for all the trials was calculated using Visual 3D (Visual 3D, C-Motion, Germantown, MD, USA). Compared to the variability of stride characteristics, joint kinematics are more sensitive for differences between groups or conditions [30]. In all trials, 3500 data points were analyzed, which is sufficient for at least 30 continuous footfalls; the recommended length to perform nonlinear analysis [31]. In each trial, the average and standard deviation of range of motion (ROM) of the ankle, knee, and hip joints were calculated. The ranges of motion were calculated using MATLAB (MathWorks Inc., Natick, MA, USA) codes based on discrete peak points (maximums and minimums in each cycle). 

The linear analysis only focuses on a few specific discrete points from the time series, while nonlinear analysis (largest Lyapunov exponent and sample entropy) considers all points in the joint angle time series. Nonlinear analysis was performed on the ankle, knee, and hip joint angle time series [32]. Sample entropy measurement can be used to assess the predictability or regularity of gait signals [33,34]. Sample entropy estimates whether future patterns will be similar to patterns earlier in the data [35]. In general, entropy values close to zero indicate highly regular systems that are repetitive, whereas larger values indicate irregular systems that have novel patterns. In this study, sample entropy for each time series was calculated to provide an evaluation of the regularity of the joint angle time series [34,35,36,37]. The parameters used included a vector length (m) of 2 and a tolerance (r) of 0.2 times the standard deviation of the time series after examining the relative consistency.

Largest Lyapunov exponents are calculated as the local divergence from an attractor (i.e., steady state condition), which quantifies the ability of the motor system to attenuate small perturbations [38]. Previous studies in patients with PAD used the largest Lyapunov exponent as a useful dynamical method to characterize variability differences from healthy controls [3,38]. The largest Lyapunov exponent approximates an individual’s ability to respond to movement disturbances or perturbations. A Lyapunov exponent value around zero has little to no divergence, and is a steady system (e.g., sine wave). A noisy system with a lot of divergence (e.g., white noise), has an exponent value around 0.5 [3]. The current study used a customized MATLAB script to calculate the largest Lyapunov exponent and sample entropy for each joint angle time series.

### 2.5. Statistics

We used Shapiro-Wilk and Kolmogorov-Smirnov tests for examining normality in addition to visually inspecting data using histograms. Normally distributed data allowed the use of two-way, repeated measure ANOVAs. The dependent variables were the mean and standard deviation of ROM, the largest Lyapunov exponent and sample entropy, while intervention and joint were the independent variables. In case any main effects (independent variables) or the interaction (intervention × joint angle) met adjusted level of significance, a Tukey post hoc test was applied to investigate the between-intervention and between-joint differences. Data were analyzed with SPSS version 23 (IBM Corp., Armonk, New York, NY, USA). The statistical significance was set at α = 0.05.

Nonsignificant results of the ANOVA analysis were contrary to our hypothesis that the Lyapunov exponent would decrease following intervention were further tested using equivalence testing. The Bayesian factor was used to measure the strength of evidence in favor of the null hypothesis, that there are no differences in Lyapunov exponent values before and after an intervention, over an alternative hypothesis. The Bayesian factor is distinct from effect sizes, which are based on the magnitude of difference in the data, rather than the likelihood of one hypothesis occurring over another. We used default prior distribution (a two-tailed Cauchy distribution) centered on zero, with a scaling factor of 0.707 [39], comparing the Lyapunov exponent before and after the intervention to calculate the effect size that is needed to determine the Bayesian factor. The Bayesian factor is used to quantify evidence in favor of the null hypothesis (B_01_) to determine whether the lack of significant differences is due to low sample size. These statistical tests were performed in JASP (Version 0.11.1).

## 3. Results

Four subjects were excluded from the final analysis due to an insufficient number of time points (~3500) in the angle time series. Therefore, data of the remaining thirty-nine subjects were included in the analysis (Table 1).

### 3.1. Nonlinear Measures

Sample entropy values decreased after the six month SET program across all joint angles (*p* = 0.008), indicating joint angles become more regular following intervention (Figure 1). Sample entropy was also significantly different between joints (*p* < 0.001), where ankle sample entropy value was greater than the knee and hip values (*p*’s < 0.001), and knee sample entropy was lower than hip sample entropy (*p* < 0.001). Overall, joint angles became more regular with proximity to the center of mass.

Average Lyapunov exponent values were not significantly different before and after the SET intervention (Figure 1, Table 2). Lyapunov exponent was significantly different across joints (*p* < 0.001). The ankle Lyapunov exponent value was greater than knee and hip (*p* < 0.001), indicating greater instability for joints as the distance from center of mass increased.

### 3.2. Linear Measures

The average values for the ranges of motion from the entire gait cycle for the ankle, knee, and hip angles before and after intervention are displayed in Figure 2. There was not a significant main effect of the intervention. There was a significant main effect of joint. The knee range of motion was greater than the ankle and hip (*p* < 0.001) and hip range of motion was greater than the ankle (*p* < 0.001) for combined pre- and post-conditions.

ROM standard deviation (Figure 2, Table 2) showed significant interactions between joint and intervention meaning that the amount of variability was quite similar before and after SET for ankle and hip joints, while it increased after intervention for the knee joint angle (*p* < 0.001).

### 3.3. Bayesian Statistics

We used Bayesian paired-sample *T*-tests for the comparison of largest Lyapunov exponent in the ankle, knee, and hip joint angles before and after SET because the results did not support our initial hypothesis. This allowed us to investigate if there was evidence supporting the null hypothesis (no significant difference of intervention), or if the lack of significant effect of the intervention was due to low sample size. We found that the B_01_ provided at least moderate evidence that the null hypothesis was true (no difference before and after intervention) relative to the alternative hypothesis for all joint comparisons (Table 3, Figure 3). The descriptive values of Lyapunov exponent values before and after intervention for three joint angles were demonstrated in Table 4. 

## 4. Discussion

This study investigated if a six-month SET intervention the affected linear and nonlinear variability of joint angles during walking. The exploration of variability has potential importance in understanding human movement [32]. In support of our initial hypothesis, sample entropy of the ankle, knee, and hip joint angles decreased after the six-month SET intervention. Contrary to our hypothesis, the largest Lyapunov exponent values were not different following the SET intervention. The linear measures, including average and standard deviation of joint ROM, did not significantly change before and after the intervention. The current study identified that applying multiple variability tools improved change detection. Previous studies using the Normalized Root Mean Square and Vector coding measures of variability also showed no differences before and after 12 months of SET [42]. However, in this study, subtle differences in the results of nonlinear measures could identify the essence of awareness about the underlying mathematical algorithm used for variability detection in varying datasets.

In support of our first hypothesis, sample entropy differed significantly before and after the SET intervention. Sample entropy was indicative of a more regular pattern of motion in the joint angle time series after SET. Alterations in gait variability reflected changes in the locomotor system [43,44]. It is probable that the increased demands placed on lower limb muscles contributed to the greater irregularity before SET intervention. After SET intervention, the distance patients with PAD could walk increased [11,45] and ankle, knee, and hip joint patterns were more regular. Ankle angles were still more irregular compared with knee and hip joints, which could be related to the impaired calf muscles in patients with PAD [46]. The hip is a ball and socket as opposed to a hinge type of a joint like a knee. There are different types of soft tissues around it, it’s a deeper joint, and there are more muscles. The greater entropy in the hip could be the results of complex structure and muscle contributions.

According to the results of the current study, the largest Lyapunov exponent values were not significantly different before and after a SET intervention when patients walked prior to the onset of claudication pain. A potential explanation for lack of significant changes could be the length of dataset, which was 3500 data points. This impacts a key parameter in the largest Lyapunov exponent calculation, the embedding dimension. The embedding dimension characterizes the complexity of the data vectors within a particular space. The largest Lyapunov exponent algorithm was fundamentally affected by the length of the time series [47]. The signal to noise ratio increased when the data length expanded. Therefore, longer length of data resulted in more accurate and consistent largest Lyapunov exponent calculations [47]. Moreover, nonlinear variability picks up on the coordination of all components of the locomotor system and this was reflected in the Lyapunov exponent values. In previous work, we observed that the largest Lyapunov exponent values were increased in patients with PAD versus healthy controls. These differences increased after the onset of claudication pain [3,10,14,19]. We also observed increased values of the largest Lyapunov exponent in healthy individuals when reduced blood flow was induced with a simulated occlusion [3,14]. A prior study examining gait changes following a SET intervention reported that pain tolerance and walking endurance improved after SET, but self-selected average walking speed and average ankle-brachial index did not improve [11]. It was also reported that although some biomechanical measures improved after SET, gait was not fully restored to the level of similar-aged controls without PAD. This may partially explain why our results did not show any significant improvement for the largest Lyapunov exponent after the SET intervention [36]. 

The sample entropy and largest Lyapunov exponent measures the structural changes across the entire joint angle time series [10]. The linear analysis examines the magnitude and amount of the variation at only specific time points within a time series [3,10]. Patients with PAD have a greater average and standard deviation of ROM in the knee joint than in the ankle and hip joints before and after intervention [10]. The amount of ankle and hip variability was similar before and after the SET intervention. However, knee joint angle linear variability increased after the SET intervention. It has been shown that joint characteristics were more sensitive compared to variability of other stride characteristics [48]. Previous studies demonstrated that patients with PAD have lower knee extensor power generation compared to healthy individuals [5,13,49,50]. Knee extensor power generation was indicative of quadriceps strength [51], which resulted in lower knee power. Increasing the magnitude of knee variability could be the effect of altered muscular function after intervention [52,53]. 

The current study has several limitations. According to our previous work [35], the minimum sampling rate that was appropriate for calculation of variability and sample entropy is equal to 120 Hz. Although the statement was based on spatiotemporal analysis of gait, it may also affect the nonlinear calculation of joint angles. Secondly, since variability was not the original focus of this study, we used the pain free condition, which quite reduced number of data points available for variability measures. In addition, future studies may focus on other conditions (e.g., maximum walking distances). This could further enhance the understanding of clinical use for training goals. We learned that SET cannot be the terminal treatment for PAD. A complementary rehabilitation or intervention (e.g., assistive device and/or strength training) may be needed to enhance walking following SET treatment to fully restore walking performance.

## 5. Conclusions

This study is the first to evaluate linear and nonlinear measures of gait variability before and after an SET intervention. The findings that gait regularity was improved, while the random structure documented in PAD gait patterns was not improved had significant clinical implications. Specifically, it demonstrated that while SET improved walking distances and some gait parameters, coordination of the locomotor system was not restored to the level seen in older individuals without PAD. Further exploration into the functional outcomes for patients with PAD after an SET program, calculating the meaningful improvement in the variability parameters using minimal clinically important difference (MCID), and companion therapies or devices including manipulating exercise dosage, rehabilitation of muscles to promote coordination, or implementing an assistive device would expand knowledge and understanding of this important topic.

## Figures and Tables

**Figure 1 entropy-24-01422-f001:**
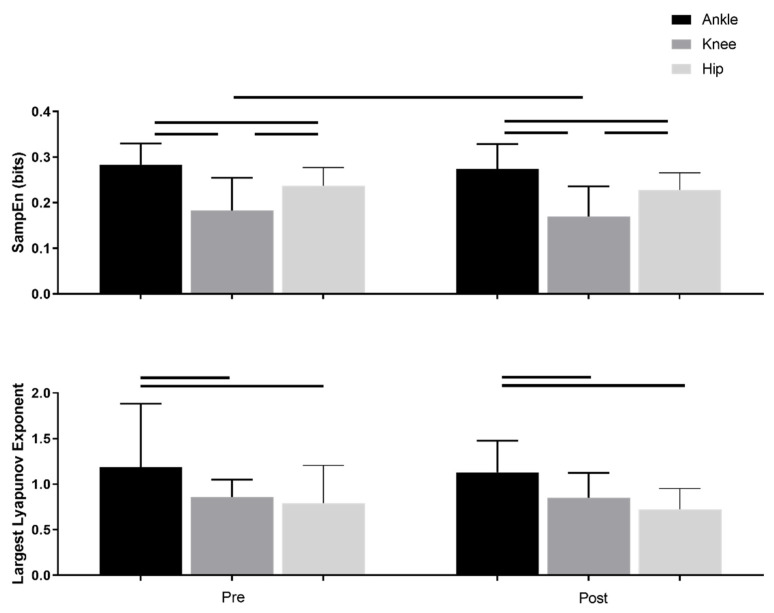
Nonlinear measures of sample entropy (**top**) and largest Lyapunov Exponent (**bottom**) of the sagittal angular motion for three lower limb joints (ankle, knee, and hip) before (Pre) and after (Post) SET. The horizontal bars represent the significant difference (*p* < 0.05).

**Figure 2 entropy-24-01422-f002:**
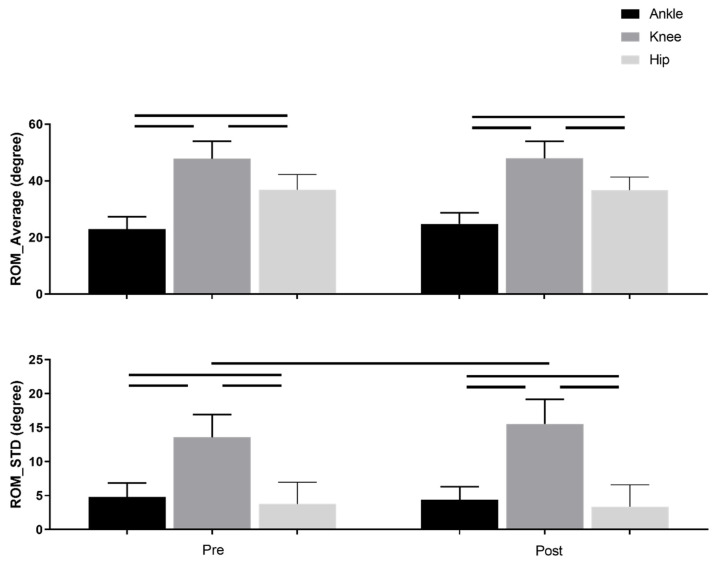
Mean ranges of motion (ROM) (**top**) and standard deviation (STD) of ranges of motion (**bottom**) for the ankle, knee, and hip before (Pre) and after (Post) SET. The horizontal bars represent the significant difference (*p* < 0.05).

**Figure 3 entropy-24-01422-f003:**
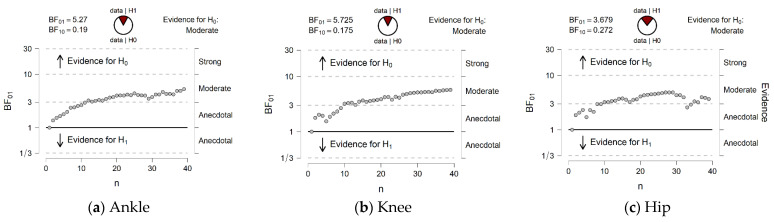
Sequential Bayesian Factor analysis was performed for Lyapunov exponent of (**a**) Ankle, (**b**) Knee, and (**c**) Hip joint angles. Sequential analysis assesses the Bayesian factor after each data point is added (sample size). The *X*-axis is the sample size in this study, and the *Y*-axis indicates the Bayesian factor in favor of the null hypothesis. The evidence labels are produced in JASP, which are based roughly on [40]. See [41] for more detail.

**Table 1 entropy-24-01422-t001:** Demographic data for the participants was used in the analysis. All values are written as mean (standard deviation).

Age (Years)	Body Mass (kg)	Height (cm)	BMI (kg/m^2^)
63.79 (6.21)	91.53 (18.6)	1.76 (0.07)	29.49 (5.7)

BMI = body mass index.

**Table 2 entropy-24-01422-t002:** Measures of sample entropy, largest Lyapunov Exponent, range of motion average and standard deviation of the sagittal joint angle motion for three lower limb joints (ankle, knee, and hip) before (Pre) and after (Post) SET. *p*-values were reported for each factor of comparison: (1) joints (i.e., ankle, knee, and hip), (2) intervention (i.e., pre, post) and their interaction.

	Ankle (Pre)	Ankle (Post)	Knee (Pre)	Knee (Post)	Hip (Pre)	Hip (Post)	*p*-Value-Joint	*p*-Value-Intervention	*p*-Value-Interaction
SampEn	0.28(0.04)	0.27(0.05)	0.18(0.7)	0.17(0.06)	0.23(0.03)	0.22(0.03)	*p* < 0.001	*p* = 0.008	*p* = 0.93
Largest Lyapunov Exponent	1.18(0.69)	1.13(0.34)	0.85(0.19)	0.85(0.27)	0.79(0.41)	0.72(0.22)	*p* < 0.001	*p* = 0.43	*p* = 0.79
ROM_Average (degree)	22.93(4.3)	24.84(3.83)	47.79(6.24)	47.92(5.98)	36.80(5.41)	36.80(4.61)	*p* < 0.001	*p* = 0.11	*p* = 0.08
ROM_STD (degree)	4.80(2.07)	4.38(1.92)	13.56(3.34)	15.55(3.59)	3.75(3.20)	3.38(3.21)	*p* < 0.001	*p* = 0.29	*p* < 0.001
*p* < 0.05 denotes significant differences between factors									

**Table 3 entropy-24-01422-t003:** Bayesian Paired Sample T-test of the ankle, knee, and hip Lyapunov exponent (LyEs) for pre- and post-intervention.

Measure 1	Measure 2	BF01	Error%
LyE_Ankle_Pre	LyE_Ankle_Post	5.27	7.03 × 10^−6^
LyE_Knee_Pre	LyE_Knee_Post	5.72	7.96 × 10^−6^
LyE_Hip_Pre	LyE_Hip_Post	3.68	4.73 × 10^−6^

**Table 4 entropy-24-01422-t004:** Descriptive statistics for Lyapunov exponent (LyE) before and after intervention.

	N	Mean	SD	SE	95% Confidence Interval	
					Lower	Upper
LyE_Ankle_Pre	39	1.185	0.698	0.112	0.959	1.411
LyE_Ankle_Post	39	1.132	0.345	0.055	1.020	1.243
LyE_Knee_Pre	39	0.859	0.192	0.031	0.797	0.921
LyE_Knee_Post	39	0.850	0.273	0.044	0.762	0.939
LyE_Hip_Pre	39	0.791	0.414	0.066	0.656	0.925
LyE_Hip_Post	39	0.726	0.226	0.036	0.653	0.799

N = number of samples, SD = standard deviation, SE = standard error.

## Data Availability

Data is contained within the article. The data presented in this study are available in the article.

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
