# Peer review of "Joint Angle Variability Is Altered in Patients with Peripheral Artery Disease after Six Months of Exercise Intervention"

_entropy, 2022, doi:10.3390/e24101422_

Round 1
Reviewer 1 Report
This study conducted a six-month, 3 times/week supervised treadmill walking training program for patients with PAD. Both linear and non-linear approaches were adopted to analyze the variability of lower extremity joint angles time series data using ROM, SampEn, and LyE. Both major and minor concerns were listed below for major revision:
Major:
1. Title: Please revise the topic slightly that reflects the research design. The topic might make the readers wondered whether this study compared two different types of training (e.g., SET vs. conventional) and following conducted a 2-way ANOVA where the INTERVENTION is one of the independent factors, and which is not exactly true.
2. Research Design & Statistical Analysis: Following above, given that this study was a pre-post research design, a one-way ANOVA with repeated measure would be enough and appropriate (where JOINT is the only independent variable).
In Results, please add ANOVA table(s) that include ROM, SampEn, and LyE. in the context, please list the values of these variables along with the p values.
Also, the reviewer cannot understand why a following or parallel analysis (i.e., the Bayesian approach) was conducted because there was no significant changes observed using ANOVA. It was noted that the non-significant outcomes remained after running the B. factor. Please choose either one (i.e., One-way repeat ANOVA vs. the Bayesian) and discussed the outcomes accordingly.
3. SET protocol: it looks like all PAD participants received different dosage of SET regarding intensity and duration of actual TM walking. How did the authors control this bias (e.g., documented walking time)?
Minor:
line 49: please add references to the 1st sentence (SET is the primary conservative tx. for PAD.)
line64-66: the cited article #16 did not studied variability vs. risk of falls. please cite the most relevant article.
line 73: (following the previous sentence) please rephrase this sentence because it sounds like hip will show more irregular movement than ankle.
line 81-82: please rephrase and re-organize the sentence.
line 90-91: rephrase this as a directional hypothesis. e.g., which outcome measures to be reduced? SampEp? LyE? or ROM?
Method:
line 105: please revise the brachil ankle index as the common form we often see- "ankle-brachil index (ABI)"
line 110: any further explanation regarding the role of supervision throughout the SET?
line 115: is there any clear cut-off point (e.g., VAS, claudication scale) as an indicator to the participant to stop?
line 142/165: list company's full name/location to where it showed initially.
line 153: please add parameters set for calculating SampEn such as length and radius.
line 169: in addition to visually inspect normality, can this be examined using Shapiro-Wilk or Kolmogorov-Smirnov test?
Results
line 212: is this the ROM-peak or mean of the entire gait cycle?
line 232: please rephrase because the null hypothesis stated here wat not what you hypothesized.
Discussion
Please discuss why SampEn(Knee) showed more regularity than that in HIp as depicted in line 200.
wtih the journal's scope of interest, please discuss Sample Entropy first and then LyE and others.
Is there any optimal zone or cut-off value of SampEn to indicate that the decreased SampEn after the SET truly mean an improved gait? (e.g., the decreased SampEn can be interpreted as their gait pattern is too predicted as a robot.)
Author Response
Thank you for your insightful comments that helped improve our paper. We have responded to all of them in the attached document.

Reviewer 2 Report
The study is of considerable clinical and scientific interest, as supervised exercise therapy has existed in orthopedics for several decades with results of considerable clinical interest. in fact, the correction of walking has been used in the prevention of degenerative joint diseases, in the prevention and treatment of muscle-tendon pain and in the post-surgical treatment for the insertion of joint prostheses.
Excellent recruitment of materials and perfect methods applied in the study. The statistical analysis used is also excellent, in fact both the two-way repeated measures ANOVA, also interesting the statistical analysis of the variables with the post-hoc Tukey HSD test are specific on multiple evaluations with tests for the scale differences between two groups, as well as analysis with SPSS version 23 (IBM Corp., Ar-175 monk, New York). To further complete the research and to underline the goodness of the work, it is interesting that the authors did not abandon or hide the insignificant results of the ANOVA analysis contrary to the hypothesis of the study, but have further tested them using the equivalence test. . Clear the graphs and confirm the results from the data obtained. This study is the first to evaluate linear and non-linear measures of gait variability before and after SET intervention, as indicated by the authors themselves. The results of considerable clinical interest have demonstrated the very purpose of the work in a clear and scientific way.
Author Response
Thank you for your positive comments about our paper.

Reviewer 3 Report
Dear Editor,
Thank you very much for the opportunity to review this manuscript. Below I indicate some relevant points about the article.
1 - Adequate abstract.
2 – The introduction makes clear the objective and which knowledge gap will be filled.
3 – The methods are reproducible and are well detailed.
4 – I suggest removing table 1 from the results and putting it in the methods, for the characterization of the sample.
5 – In the discussion I suggest that the authors pay attention to statements that do not have a theoretical framework. Examples: “Improvements in gait regularity reflect changes in the locomotor system.” – lines 295 and 286
“Previous studies demonstrated that patients with PAD have lower knee extensor power generation compared to healthy individuals.” – Lines 300 and 301.
These are my thoughts on the manuscript. I'm available if needed.
Sincerely.
Author Response
Thank you for your supportive comments and helpful suggestions on our paper. We have addressed them in the attached document.
